# The Influence of Emotional Regulation and Cognitive Flexibility on Sleep Habits in Spanish Children and Adolescents through the Lens of Parents

**DOI:** 10.3390/children10081390

**Published:** 2023-08-15

**Authors:** Borja Costa-López, Rocío Lavigne-Cerván, Joshua A. Collado-Valero, Rocío Juárez-Ruiz de Mier, Ignasi Navarro-Soria

**Affiliations:** 1Department of Health Psychology, University of Alicante, 03690 Alicante, Spain; borja.costa@ua.es; 2Department of Developmental and Educational Psychology, University of Malaga, 29016 Málaga, Spain; rlc@uma.es (R.L.-C.); joshuaeducacion@gmail.com (J.A.C.-V.); rjrm@uma.es (R.J.-R.d.M.); 3Department of Developmental and Educational Psychology, University of Alicante, 03690 Alicante, Spain

**Keywords:** emotional regulation, sleep habits, anxiety, children, adolescents

## Abstract

Background: Previous research studies have suggested the importance of studying the relationship between emotional regulation and sleep habits. Some investigations have especially focused on how emotional regulation could impact sleep habits in children and adolescents. Therefore, these researchers have stated there exists a two-way direction in this relationship. Objective: This study aimed to analyze the influence of emotional regulation on sleep habits in Spanish children and adolescents and the mediating role of anxiety in this relationship. Method: Participants were 953 Spanish parents who completed the assessment protocol according to their children and adolescents’ information. Results: The results revealed moderate–strong correlations between emotional regulation problems and sleep habit disturbances (r = 0.375, *p* < 0.001), trait (r = 0.488, *p* < 0.001) anxiety, and state (r = 0.589, *p* < 0.001) anxiety. Additionally, emotional regulation showed a direct impact on sleep habits (β = 0.011, *p* = 0.005). Trait and state anxiety demonstrated a significant mediating role in the relationship between emotional regulation and sleep habits. Conclusions: Emotional regulation may have an impact on sleep habits during childhood and adolescence, suggesting the importance of early intervention focused on the emotions management and the prevention of sleep habit disturbances.

## 1. Introduction

Childhood and adolescence are developmental stages in which health habits and the regulation of emotional well-being are essential for an appropriate balance of mental health [1,2]. Health habits in children and adolescents are considered the set of learned behaviors that provide physical, cognitive, and emotional wellbeing to the individual once they become habits [3]. Some authors have therefore suggested that these health habits are closely related to emotional regulation, which is understood as the higher cognitive mechanisms and processes, mainly executive, that we activate whenever an emotional response arises [4,5,6,7].

Previous studies state sleep as one of the main indicators of health habits and its association with cognitive–emotional variables [8,9]. Sleep habits seem to play an important role in the proper maturation of the brain during childhood and early adolescence [10], and several studies have stated the relevance of the connection between the modulation of some cognitive processes and sleep disturbances [11,12,13,14,15]. These cognitive processes take part in executive functions (EFs), which can be influenced by changes in sleep habits, causing some attentional and memory difficulties, mood fluctuations, behavioral problems, and decreased school performance in youngsters [16]. Moreover, upon the outbreak of the COVID-19 pandemic, the number of children and adolescents presenting emotional and behavioral problems has significantly increased [17]. These emotional and behavioral problems in youngsters tend to appear since they are six years old, and they are associated with difficulties in emotion regulation strategies [18].

Emotional regulation and cognitive flexibility are also the dimensions of EFs that have been most commonly related to sleep [19,20,21,22,23,24,25]. Impairments in executive functions are related to emotional and behavioral problems, and this may contribute to the maintenance of emotional disorders [26].

Previous research has provided evidence of the association between sleep disorders and dimensions of executive functions, such as emotional regulation [27,28,29] and cognitive flexibility [30,31], with anxiety [32,33,34,35,36,37]. These findings suggest a complex relationship between these factors, where the level of development in executive functions or executive disorders can play a triggering role in the development of anxiety [27,28,29,30,31,32,37,38,39,40,41,42]. This connection between anxiety, sleep disorders, and executive functions is particularly relevant in the context of children and adolescents [41,42].

Executive functions, which include emotional regulation and cognitive flexibility, are crucial for effectively controlling and managing cognitive and emotional processes. On the other hand, anxiety can significantly impact the quality of sleep in youth [41,42]. The stress and worry associated with anxiety can lead to difficulties in falling asleep, staying asleep, or experiencing restorative sleep. Additionally, sleep disorders can negatively affect executive functions, leading to attention, memory, and emotional self-regulation difficulties in children and adolescents [16]. This vicious cycle between anxiety, sleep disorders, and executive functions can have a significant impact on mental health and well-being in youth.

Current research has demonstrated that the associations between anxiety, sleep disorders and executive functions are tricky. It is not a simple one-way relationship. Indeed, there is a bidirectional influence between these variables. For instance, sleep disorders can increase anxiety levels, while anxiety can also negatively impact sleep and executive functions. Therefore, it is essential to address these aspects comprehensively in the study and intervention of mental health in children and adolescents.

Based on recent evidence, we could therefore state that there is an interrelationship between sleep disorders, executive functions, and anxiety in children and adolescents. This complex relationship suggests that addressing these aspects comprehensively may be fundamental for promoting a proper balance of mental health in this population. The findings from these studies may have significant implications for developing interventions and strategies to improve sleep habits, emotional regulation, and cognitive flexibility, as well as reducing anxiety levels in children and adolescents. However, further studies are needed to fully understand the nature and mechanism of these associations and potentially enhance the quality of life for young individuals in the context of mental health.

Regarding the assessment of psychological aspects in children and adolescents, children’s reports tend to differ from parents’ ones [43]. In fact, studies show an overestimation of the parents when rating children’s behaviors, resulting in assessment biases [44]. However, researchers have suggested that parent proxy forms are an excellent candidate for accurately detecting cognitive and emotional aspects in youngsters [45].

Many recent researchers have focused their interest on exploring how sleep problems negatively influence emotional regulation and cognitive flexibility [46,47]. Some of them have focused their attention on investigating the bidirectional correlation between sleep habits and emotional regulation [46,47], whereas others have observed the influence of emotional regulation on health habits such as sleep [48,49]. We hypothesized that the alteration of the processes and functions of the executive system might be a predictor of sleep habits, and vice versa, and anxiety as a mediator of this relationship. Given the aforementioned, the current study aimed: (1) to analyze the correlations among sleep habits, anxiety and executive functioning, including its dimensions (emotional regulation and cognitive flexibility), (2) to determine the bidirectional influence of emotional regulation and cognitive flexibility on sleep habits; and (3) to explore the mediating role of anxiety between executive functioning and sleep habits.

## 2. Materials and Methods

### 2.1. Study Population and Design

The present research was a cross-sectional, correlational, and explicative study. The inclusion criteria for participants were: (i) parents of children between 6 and 18 years old; (ii) residing in Spain during the study period; (iii) parents aged 18 years of age or older; and (iv) adequate understanding of how to complete the evaluation protocol. Parents with sensory, physical, or psychological issues that hindered their ability to comprehend and fill out the evaluation were excluded.

The initial group of participants included 953 children and adolescents, out of which 953 (512 males and 441 females), ranging in age from 6 to 18 years (mean = 10.85; SD = 3.29), ultimately took part. This sample was drawn from individuals who responded to telematic questionnaires and were residing in Spain. The involvement of legal guardians was necessary; they completed the questionnaires on behalf of the children. Among these participants, 804 were females and 149 were males, with ages ranging from 19 to 68 years (mean = 43.30; SD = 6.70). All legal guardians were well-informed about the study’s different stages and characteristics. They provided informed consent and fulfilled the questionnaire requirements. Respondents above 18 years of age were allowed to independently provide informed consent and complete the questionnaire. Individuals who did not fully respond to the questionnaire or provide informed consent were not included in the study. The majority of respondents were of Spanish nationality (95.9%). As for the relationship these individuals had with the child or adolescent, 84.6% were maternal, 12.9% were paternal, 1.5% were siblings, and the remaining 1% comprised other relationships such as grandparents, uncles, neighbors, and/or guardians.

### 2.2. Instruments

Recipients rated their children’s anxiety using the Spanish version of the State-Trait Anxiety Inventory for Children (STAIC) test [50,51]. This test also demonstrated adequate psychometric properties [52]. The STAIC was composed of 40 items that were equally divided in two main dimensions: trait and state anxiety. The state anxiety scale tried to clarify “how the child feels at a given moment”. The trait anxiety scale measured “how the child feels in general”, exploring relatively stable differences in propensity to anxiety. Parents answered the questionnaire with their children’s information for the current study. Although the original instrument was a self-report type, parents filled the online surveys out in collaboration with children when necessary. Items were modified by including “Your child…” at the beginning of each item. For instance, “your child feels calm”, and “your child feels restless”. Response options were 7-point Likert scales, in which 1 = “strongly disagree” and 7 = “strongly agree”. For this study, the reliability of the instrument was adequate, presenting values of α = 0.894 for trait anxiety and α = 0.907 for state anxiety.

The Screening for Sleep Disorders in Childhood (BEARS) was used as a brief sleep habit disturbances screening test with 9 items [53]. This test was completed by parents/guardians answering questions such as “their child seems to be tired or drowsy” or “their child wakes up several times during the night”. Each item had 7 response options, in which 1 = totally disagree and 7 = totally agree. Regarding reliability and internal consistency, the scale presented a Cronbach’s alpha coefficient of 0.732 [54].

The assessment of executive behavior utilized the Behavioral Evaluation of Executive Function (BRIEF-2) test, specifically employing the parent-report form [55]. This instrument was comprised of nine scales, encompassing a total of sixty-three items, each offering three potential response choices: always, sometimes, or never. Among these domains, a selection was made of ten items from the emotional regulation subscale, which encompassed dimensions of emotional control and cognitive flexibility. The decision to perform a concise screening led to the choice of these specific parent-version items due to their direct relevance to the current study’s objectives. Out of these, six items were derived from emotional control, while four items pertained to cognitive flexibility. Within the present sample, these chosen items exhibited satisfactory reliability, as demonstrated by both Cronbach’s alpha and McDonald’s omega coefficients: α = 0.902, ω = 0.901 for the emotional regulation subscale, α = 0.877, ω = 0.881 for emotional control, and α = 0.832, ω = 0.830 for cognitive flexibility.

### 2.3. Procedure

The ethics committee of the University of Alicante granted approval for the present study. Families were provided with information regarding the study’s objectives, and researchers communicated that involvement was entirely confidential, anonymous, and voluntary. Parents who consented to take part were provided with a link to access the assessment protocol set up on the Google Form platform. Recruitment for participation in the study was carried out through social media groups in the months of May and June 2022, utilizing a snowball sampling approach. To ensure the protection of data confidentiality and anonymity, unique codes were assigned to individuals as part of a pseudonymization procedure. The research adhered to the principles outlined in the Declaration of Helsinki and the standards of the European Union’s Good Clinical Practice.

### 2.4. Data Analysis Plan

Preliminary analyses. Prior to conducting the primary statistical analyses in line with the study’s aims, psychometric characteristics of the Spanish modified version for parents of the STAIC were calculated for its use in parents as observers. Additionally, descriptive statistics (i.e., means, standard deviations, skewness, and kurtosis) of participants’ trait/state anxiety, sleep habits, executive functions, emotional regulation and cognitive flexibility. To examine bivariate correlations among trait/state anxiety, sleep habits, executive functions, emotional regulation, and cognitive flexibility, a correlation matrix was created. According to Hernández-Lalinde et al. [56], the interpretation of the Pearson correlation coefficient (PCC) was: 0.00 < r < 0.10 for null correlations; 0.11 < r < 0.30 for weak correlations; 0.31 < r < 0.50 for moderate correlations; and 0.51 < r < 1.00 for strong correlations.

Hierarchical regression analyses. Four hierarchical regressions were run to examine the role of trait/state anxiety and executive functions variables as predictors of sleep habit disturbances. In Model 1, trait anxiety was included as a predictor. Model 2 included trait and state anxiety as predictors. Model 3 then included trait and state anxiety and cognitive rigidity. Additionally, Model 4 included trait/state anxiety, cognitive rigidity, and emotional regulation as predictors.

Mediational models. Figure 1 shows four mediational models, which were developed using the PROCESS macro [57] to examine the direct and indirect effects of emotional regulation and cognitive flexibility on sleep habits, using 5000 bootstrap samples. Bootstrapping is a non-parametric method for assessing indirect effects [58,59]. Bootstrapping provides the most powerful and reasonable method of obtaining confidence limits for specific indirect effects under most conditions [60]. In the first two models, emotional regulation (Model 1) and cognitive flexibility (Model 2) were specified to lead to trait anxiety, which was then specified to lead to sleep habits. In the other two models, emotional regulation (Model 3) and cognitive flexibility (Model 4) were specified to lead to state anxiety, which was then specified to lead to sleep habits. Additionally, the other four models were analyzed to confirm the bidirectional influence of the study variables through mediations, in which sleep habits were introduced as predictors and emotional regulation/cognitive flexibility as outcome variables.

Analyses were performed using SPSS version 28.0 (SPSS, Inc., Chicago, IL, USA). The level of significance was set at *p* < 0.05.

## 3. Results

### 3.1. Reliability of the Spanish Modified Version for Parents of the STAIC

All of the corrected item-total correlations are above 0.30 (Table 1). The overall internal consistency of the STAIC was excellent (α = 0.936, ω = 0.938). Regarding the dimensions, State Anxiety shows the highest reliability score (Table 1).

### 3.2. Descriptive and Correlation Analysis between Trait and State Anxiety, Executive Functions, and Sleep Disturbances

Table 2 shows the descriptive analysis results and the relationship among the study variables. Specifically, both state and trait anxiety demonstrate a positive, moderate, and significant correlation with sleep habit disturbances. Moreover, state and trait anxiety are importantly associated with executive dysfunctions, especially with emotional dysregulation and cognitive rigidity. Sleep habit disturbances have also pointed out positive, moderate, and significant correlations with the executive dysfunctions global index, emotional dysregulation, and cognitive rigidity (Table 2).

### 3.3. State/Trait Anxiety and Executive Dysfunctions as Predictors of Sleep Habit Disturbances

Table 3 shows the regression models in which trait/state anxiety, emotional dysregulation, and cognitive rigidity are considered predictors of sleep habit disturbances. Although cognitive rigidity appears to be a significant predictor of sleep habit disturbances, that variable is not significant when emotional dysregulation is introduced in the regression model. Trait/state anxiety remains a significant predictor of sleep habit disturbances in all models.

### 3.4. The Mediating Role of the Anxiety between Executive Functions and Sleep Disturbances

Four mediational models were built to explore the mediation effects of anxiety in the relationship between executive functioning dimensions (emotional regulation and cognitive flexibility) and sleep habits, controlling children’s age, parents’ marital status, and parents’ educational level. Emotional regulation and cognitive flexibility have been entered into the model as independent variables, trait/state anxiety as mediators, and sleep habits have been evaluated as dependent variables.

Figure 2 and Figure 3 show the mediational effects of trait/state anxiety, in which emotional regulation predicts sleep habits (B = 0.143, SE = 0.013, 95% CI [0.03–0.08], *p* < 0.001), trait anxiety (B = 0.481, SE = 0.021, 95% CI [0.32–0.40], *p* < 0.001), and state anxiety (B = 0.529, SE = 0.024, 95% CI [0.41–0.51], *p* < 0.001). Regarding the mediation variables, sleep habits are predicted by trait (B = 0.224, SE = 0.019, 95% CI [0.07–0.15], *p* < 0.001) and state (B = 0.199, SE = 0.017, 95% CI [0.05–0.12], *p* < 0.001) anxiety. Moreover, the analyses of the indirect effect of trait and state anxiety show significant mediations (indirect effect of trait anxiety: B = 0.107, SE = 0.019, 95% CI [0.07–0.15], *p* < 0.001); indirect effect of state anxiety: B = 0.106, SE = 0.021, 95% CI [0.07–0.15], *p* < 0.001).

Finally, in relation to the other mediational models, cognitive flexibility predicts sleep habits (B = 0.190, SE = 0.019, 95% CI [0.15–0.23], *p* < 0.001), trait anxiety (B = 0.473, SE = 0.037, 95% CI [0.40–0.54], *p* < 0.001), and state anxiety (B = 0.668, SE = 0.041, 95% CI [0.59–0.75], *p* < 0.001). Regarding the mediation variables, sleep habits are predicted by trait (B = 0.213, SE = 0.015, 95% CI [0.18–0.24], *p* < 0.001) and state (B = 0.178, SE = 0.013, 95% CI [0.15–0.20], *p* < 0.001) anxiety. Moreover, the analysis of the indirect effect of trait and state anxiety shows significant mediations (indirect effect of trait anxiety: B = 0.082, SE = 0.01, 95% CI [0.06–0.10], *p* < 0.001); indirect effect of state anxiety: B = 0.096, SE = 0.012, 95% CI [0.07–0.12], *p* < 0.001) (see Figure 4 and Figure 5).

Other four mediational models were examined to confirm the bidirectional relationship between the study variables. Thus, Figure 6 and Figure 7 represent the effects of sleep habits on emotional regulation and cognitive flexibility mediated by trait anxiety. Sleep habits seem to predict trait anxiety (B = 0.867, SE = 0.06, 95% CI [0.75–0.99], *p* < 0.001), emotional regulation (B = 0.510, SE = 0.084, 95% CI [0.35–0.68], *p* < 0.001), and cognitive flexibility (B = 0.304, SE = 0.06, 95% CI [0.19–0.42], *p* < 0.001). The analyses of the indirect effect of trait anxiety in the relationship between sleep habits and emotional regulation show a significant mediation (B = 0.465, SE = 0.05, 95% CI [0.38–0.56], *p* < 0.001), and also between sleep habits and cognitive flexibility (B = 0.262, SE = 0.03, 95% CI [0.20–0.32], *p* < 0.001).

Figure 8 and Figure 9 represent the effects of sleep habits on emotional regulation and cognitive flexibility mediated by state anxiety. Figure 8 and Figure 9 indicate the effects of sleep habits on emotional regulation and cognitive flexibility, mediated by state anxiety. Sleep habits seem to predict state anxiety (B = 1.006, SE = 0.07, 95% CI [0.87–1.14], *p* < 0.001), emotional regulation (B = 0.444, SE = 0.08, 95% CI [0.28–0.61], *p* < 0.001), and cognitive flexibility (B = 0.236, SE = 0.06, 95% CI [0.13–0.35], *p* < 0.001). The analyses of the indirect effect of state anxiety in the relationship between sleep habits and emotional regulation show a significant mediation (B = 0.532, SE = 0.05, 95% CI [0.44–0.63], *p* < 0.001), as well as between sleep habits and cognitive flexibility (B = 0.329, SE = 0.03, 95% CI [0.27–0.39], *p* < 0.001).

## 4. Discussion

Sleep and executive functions are influenced by child development factors. Several researchers have assessed the effect of sleep on EFs, especially the management of emotions. Thus, a great number of researches related to anxiety and restorative sleep, which conclude in a deterioration of these variables and, therefore, of mental health in childhood and adolescence, have been developed so far [61,62,63,64]. Among these investigations, some of them indicate different protective mediational factors between anxiety and sleep, such as executive functions that allow us to carry out adequate emotional management [34,35,65,66]. However, there is a scarce number of studies focusing on the impact of emotional regulation on sleep habits. The present study aimed to analyze the influence of EFs (emotional regulation and cognitive flexibility) on sleep habits while considering the mediating role of trait/state anxiety in children and adolescents through the parents’ perception. We therefore set the following objectives: (1) analyzing the relationship among sleep habits, trait/state anxiety, and EFs (emotional regulation and cognitive flexibility); (2) exploring the influence of the trait/state anxiety, emotional regulation, and cognitive flexibility on the sleep habits; and (3) examining the mediating role of state/trait anxiety in the relationship between the EFs (emotional regulation and cognitive flexibility) and sleep habits.

In respect of the relationship among the variables measured, the findings revealed moderate–strong and positive correlations. That is, the higher the trait/state anxiety, the higher the sleep habit disturbances and alterations in EFs. These results are consistent with previous research studies, in which they highlighted that the increasing level of anxiety could be related to the appearance of sleep habit disturbances [41,42,67]. The relationship between these two variables has been known for a long time. In fact, some studies have suggested that anxiety and sleep are strictly related and affect each other in a two-way manner [68]. Moreover, other recent studies have stated the direct association of the levels of anxiety with alterations in emotional regulation and cognitive flexibility [33,34,35,36]. Further, based on the results of previous scientific studies, sleep habit disturbances seem to be linked to a higher level of difficulty in controlling EFs, especially emotional regulation and cognitive flexibility [20,21,22,69].

Regarding the findings observed in the multiple regressions, trait/state anxiety appears to be significant in all models. This is coherent with other investigations, in which they identified anxiety as a predictor of the fluctuation of sleep habits in children and adolescents [41,70]. Moreover, emotional regulation also showed an influence on sleep habits. As a matter of fact, recent literature has suggested further consideration of the management of emotions to fully understand sleep habits [22,47]. Furthermore, the results have demonstrated the importance of the mediating role of anxiety in the relationship between EFs (emotional regulation and cognitive flexibility) and sleep habits. These findings are also consistent with other research studies in which anxiety played an essential role in the measured relationship [34,35]. Previous studies from different countries have supported the association between sleep and emotional problems in children [71]. It has also been indicated that children may present different sleep problems associated with emotional problems [72,73].

## 5. Strength, Limitations, and Future Research

The current study’s findings could help the scientific and clinical community further understand the need to examine the mechanisms associated with the relationship between emotional regulation and sleep habits. These results also allow professionals and the rest of society to identify and prevent sleep disturbances related to maladaptive emotional strategies. However, although this research comprises a great sample of participants and rich data analysis, which let us explain the performance of the variables measured, it is not without limitations that we are aware of. This study was carried out from the perspective of parents. As the questionnaires were sent via social networks to parents, despite parental observation being known as one the most reliable methods to assess children’s behaviors, this could be a reason for some biases in the results [65,66]. Measures are vulnerable to recall bias, and findings should be therefore interpreted with caution. In particular, for this research, we used the Spanish modified version of the STAIC. This questionnaire was originally designed for a self-report version in children, which is why it could cause some results biases, as well as difficulties in the generalization of the findings. For this reason, it is essential to approach them with caution, as previously stated.

In regard to the BEARS, although it is a suitable data source for sleep habits that contains some validated measures, it does not include all the dimensions we would have liked to explore. For example, information on sleep quality, such as subjective sleep quality or possible pharmacological treatments for altered sleep quality, was not available. This could differentiate the sleep habits and quality.

As for future research lines, it is essential to carry out investigations in which children are the direct participants. In fact, including self-rating data in youth is essential, and this could avoid some difficulties in exploring their behavior and parents’ biases [74]. Families, especially parents, are more likely to have the ability to more accurately rate externalizing behaviors than internalizing ones. Children’s self-reports present more sensitivity when assessing internalizing behaviors [75]. Adding more informants could also enrich the assessment of children’s mental health [76]. Indeed, even though it could result in biases in the assessment of internalizing behaviors in children, the Spanish modified version for parents of the STAIC could offer the opportunity to adapt and validate this new version for other respondents (parents or teachers). Additionally, this may encourage other researchers to employ this modified version since it appears to present good psychometric properties.

Additionally, analyzing the similarities and differences between sex or age groups would provide the scientific community with interesting data about the manifestation of the variables measured between males and females and also across ages. Further, another crucial issue to deal with is to explore other protection and vulnerability factors of sleep habits through a latent variable analysis, in which other psychosocial factors could be considered to better understand children and adolescents’ behavior.

## 6. Conclusions

The present study sheds light on the significance of employing effective emotional regulation strategies and managing anxiety in order to promote healthy sleep habits. It is evident that higher levels of anxiety are associated with increased disruptions in the sleep patterns of children and adolescents, with anxiety and emotional regulation emerging as influential factors, as reported by parents. Furthermore, emotional regulation and cognitive flexibility can be considered protective factors for overall health habits, including sleep. In light of these findings, it is imperative for healthcare professionals to prioritize the identification and evaluation of potential sleep-related issues in children and adolescents. This is particularly crucial as anxiety-inducing environments, such as school, family, and work settings, can have detrimental effects on sleep habits and executive functioning. Therefore, it becomes essential to enhance the utilization of emotional and cognitive strategies to facilitate the regulation of sleep habits effectively.

It is important to recognize that the promotion of adequate sleep habits is a multidimensional task that requires a comprehensive approach. By addressing emotional regulation and cognitive flexibility, professionals can help individuals develop the necessary skills to regulate their sleep patterns more effectively. This includes the identification of triggers that may disrupt sleep, the implementation of relaxation techniques, and the establishment of consistent bedtime routines. Additionally, providing education and support to parents and caregivers is vital in ensuring a conducive sleep environment and reinforcing positive sleep habits.

By integrating emotional and cognitive strategies into sleep interventions, the overall well-being and executive performance of children and adolescents can be significantly improved. The recognition of the intricate relationship between emotions, anxiety, and sleep habits underscores the need for a holistic approach that considers the interplay of various factors. Ultimately, fostering healthy sleep habits early on can have long-lasting benefits, contributing to optimal development and overall quality of life.

## Figures and Tables

**Figure 1 children-10-01390-f001:**
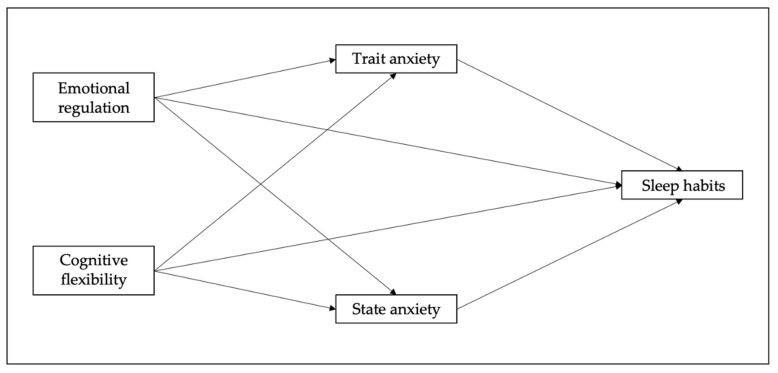
Research mediational models of the present study, showing the hypothesized effects of emotional regulation/cognitive flexibility on sleep habits.

**Figure 2 children-10-01390-f002:**
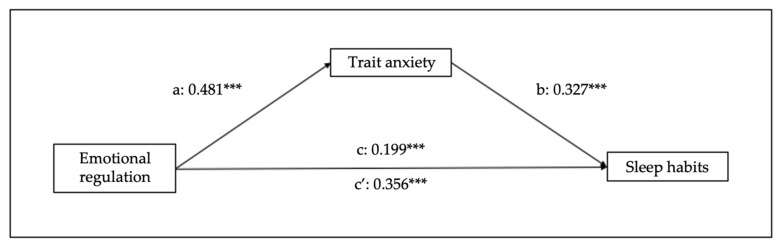
Results of regression analysis for the mediation effects of emotional regulation on sleep habits mediated by trait anxiety. a, b, c, c’ represent standardized regression coefficients. *** *p* < 0.001.

**Figure 3 children-10-01390-f003:**
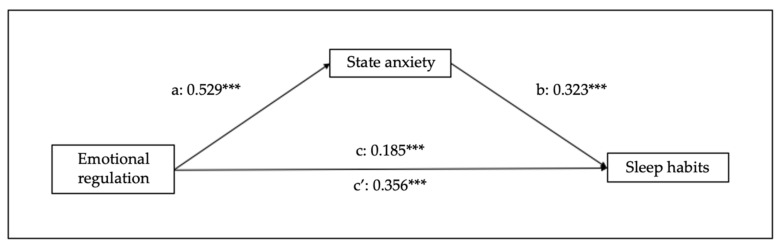
Results of regression analysis for the mediation effects of emotional regulation on sleep habits mediated by state anxiety. a, b, c, c’ represent standardized regression coefficients. *** *p* < 0.001.

**Figure 4 children-10-01390-f004:**
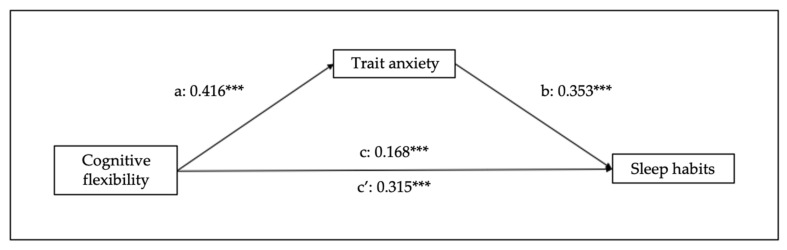
Results of regression analysis for the mediation effects of cognitive flexibility on sleep habits mediated by trait anxiety. a, b, c, c’ represent standardized regression coefficients. *** *p* < 0.001.

**Figure 5 children-10-01390-f005:**
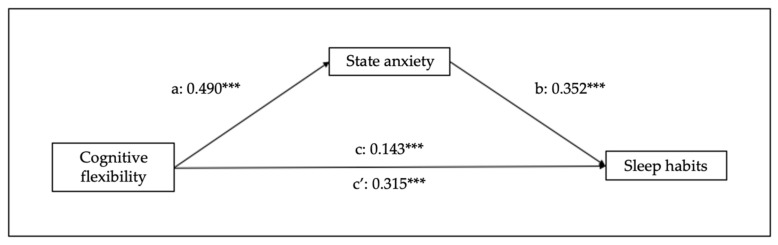
Results of regression analysis for the mediation effects of cognitive flexibility on sleep habits mediated by state anxiety. a, b, c, c’ represent standardized regression coefficients. *** *p* < 0.001.

**Figure 6 children-10-01390-f006:**
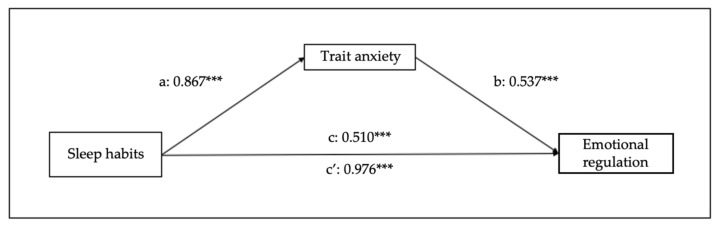
Results of regression analysis for the mediation effects of sleep habits on emotional regulation mediated by trait anxiety. a, b, c, c’ represent standardized regression coefficients. *** *p* < 0.001.

**Figure 7 children-10-01390-f007:**
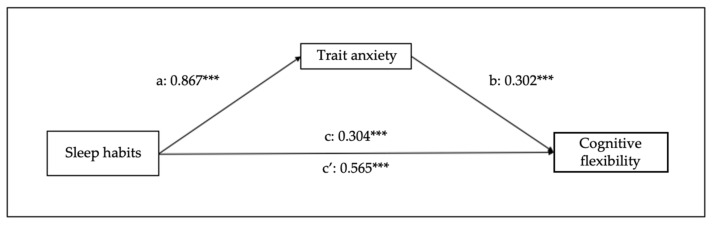
Results of regression analysis for the mediation effects of sleep habits on cognitive flexibility mediated by trait anxiety. a, b, c, c’ represent standardized regression coefficients. *** *p* < 0.001.

**Figure 8 children-10-01390-f008:**
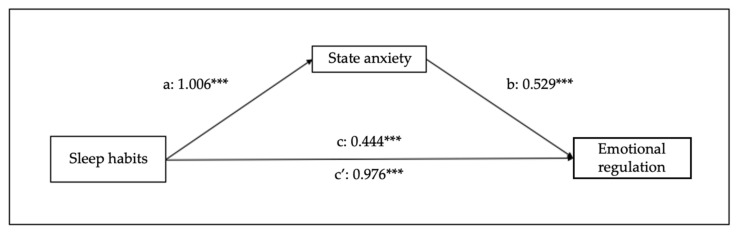
Results of regression analysis for the mediation effects of sleep habits on emotional regulation mediated by state anxiety. a, b, c, c’ represent standardized regression coefficients. *** *p* < 0.001.

**Figure 9 children-10-01390-f009:**
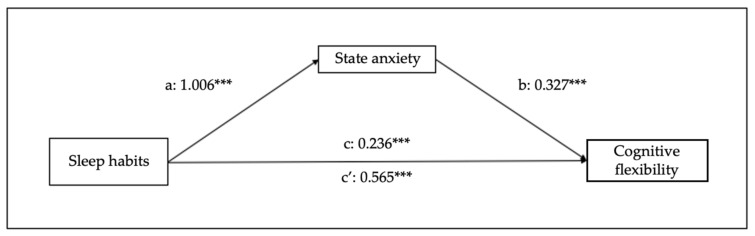
Results of regression analysis for the mediation effects of sleep habits on cognitive flexibility mediated by state anxiety. a, b, c, c’ represent standardized regression coefficients. *** *p* < 0.001.

**Table 1 children-10-01390-t001:** Psychometric characteristics of the Spanish modified version of the STAIC for parents.

Item	r_it_^c^	α-i	ω-i
Trait Anxiety			
1	0.576	0.887	0.897
2	0.544	0.888	0.899
3	0.602	0.886	0.897
4	0.375	0.893	0.902
5	0.543	0.888	0.898
6	0.542	0.888	0.898
7	0.521	0.889	0.899
8	0.552	0.888	0.898
9	0.593	0.887	0.898
10	0.513	0.889	0.899
11	0.583	0.888	0.898
12	0.606	0.886	0.896
13	0.650	0.885	0.896
14	0.582	0.887	0.897
15	0.618	0.886	0.895
16	0.553	0.916	0.916
17	0.679	0.884	0.893
18	0.602	0.887	0.898
19	0.599	0.886	0.896
20	0.655	0.885	0.895
Total reliability score of the subscale: α = 0.894, ω = 0.903			
State Anxiety			
1	0.400	0.907	0.908
2	0.560	0.903	0.904
3	0.553	0.903	0.904
4	0.524	0.903	0.905
5	0.537	0.903	0.904
6	0.611	0.901	0.902
7	0.607	0.902	0.903
8	0.654	0.900	0.901
9	0.300	0.909	0.910
10	0.519	0.904	0.905
11	0.553	0.903	0.904
12	0.604	0.901	0.902
13	0.573	0.902	0.903
14	0.414	0.907	0.908
15	0.524	0.903	0.904
16	0.563	0.902	0.903
17	0.716	0.898	0.899
18	0.583	0.902	0.903
19	0.624	0.901	0.902
20	0.576	0.902	0.904
Total reliability score of the subscale: α = 0.907, ω = 0.908			

Note. r_it_^c^ = correlation of item-total test, *α*-i = reliability if the item is dropped, *ω*-i = reliability if the item is dropped.

**Table 2 children-10-01390-t002:** Means, standard deviations, skewness, kurtosis, and correlations (confidence intervals) among the study variables.

Variables	M (SD)	Skw	Kurt	1.	2.	3.	4.	5.	6.
1. Trait anxiety	33.57 (7.49)	0.34	−0.40	-					
2. State anxiety	34.99 (8.72)	0.48	−0.37	0.654 *** [0.616, 0.689]	-				
3. Sleep habit disturbances	13.17 (3.65)	0.74	−0.14	0.423 *** [0.369, 0.474]	0.422 *** [0.368, 0.472]	-			
4. Executive dysfunctioning global index	41.84 (16.63)	0.21	−0.85	0.488 *** [0.438, 0.534]	0.559 *** [0.514, 0.601]	0.375 *** [0.319, 0.428]	-		
5. Emotional control problems	21.86 (10.01)	0.16	−1.09	0.481 *** [0.430, 0.528]	0.529 *** [0.481, 0.573]	0.356 *** [0.299, 0.410]	0.921 *** [0.911, 0.930]	-	
6. Cognitive rigidity	13.49 (6.55)	0.35	−0.85	0.416 *** [0.362, 0.467]	0.491 *** [0.441, 0.537]	0.315 *** [0.257, 0.371]	0.866 *** [0.849, 0.881]	0.653 *** [0.615, 0.688]	-

Note. M = Mean, SD = Standard deviation, Skw = Skewness, Kurt = Kurtosis. *** Correlation is significant at the 0.001 level (2-tailed).

**Table 3 children-10-01390-t003:** Regression analyses of sociodemographic, state/trait anxiety, and executive functioning variables as predictors of sleep habit disturbances.

Variable	Sleep Habit Disturbances				
	Model 1				
	B	SE	β	*p*	95% CI
Trait anxiety	0.203	0.015	0.415	<0.001	[0.357, 0.473]
	F(1) = 188.50, R^2^ = 0.187, ∆R^2^ = 0.167, *p* < 0.001				
	Model 2				
	B	SE	β	*p*	95% CI
Trait anxiety	0.128	0.019	0.263	<0.001	[0.188, 0.337]
State anxiety	0.100	0.017	0.238	<0.001	[0.163, 0.313]
	F(2) = 116.18, R^2^ = 0.219, ∆R^2^ = 0.032, *p* < 0.001				
	Model 3				
	B	SE	β	*p*	95% CI
Trait anxiety	0.120	0.019	0.246	<0.001	[0.171, 0.321]
State anxiety	0.079	0.017	0.189	<0.001	[0.110, 0.268]
Cognitive rigidity	0.048	0.020	0.087	0.015	[0.017, 0.156]
	F(3) = 84.06, R^2^ = 0.230, ∆R^2^ = 0.005, *p* = 0.015				
	Model 4				
	B	SE	β	*p*	95% CI
Trait anxiety	0.112	0.019	0.229	<0.001	[0.153, 0.304]
State anxiety	0.071	0.017	0.168	<0.001	[0.088, 0.249]
Cognitive rigidity	0.019	0.022	0.034	0.403	[−0.045, 0.113]
Emotional control problems	0.041	0.015	0.112	0.006	[0.033, 0.192]
	F(4) = 65.50, R^2^ = 0.236, ∆R^2^ = 0.006, *p* = 0.006				

Note. Significant differences were considered when *p* < 0.05.

## Data Availability

All data generated or analyzed during this study are included in this published article. The availability of data must be personally requested to the corresponding author at ignasi.navarro@ua.es.

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
