# Peer review of "The Influence of Emotional Regulation and Cognitive Flexibility on Sleep Habits in Spanish Children and Adolescents through the Lens of Parents"

_children, 2023, doi:10.3390/children10081390_

Round 1

Reviewer 1 Report

Comment

This study examined an important health issue. The results of this study provided knowledge to the filed of sleep health in children and adolescents.

I have some suggestions for the authors to improve this manuscript.

1.      “Cognitive flexibility” was one of variables examined in this study; however, it did not appear in the title.

2.      The authors mentioned “the bidirectional correlation between sleep habits and emotional regulation.” The pathway of emotional regulation to sleep problems via anxiety was examined in this study. Why did the authors not consider the pathway of sleep problems to emotional regulation/cognitive flexibility via anxiety? According to the neuropsychological mechanism, this pathway can be hypothesized.

3.      The reasons to examine the mediating role of anxiety for the association between emotional regulation/cognitive flexibility and sleep problems should be introduced.

4.      Sleep disturbance is often one of surveyed item in the instruments for anxiety. If the State-Trait Anxiety Inventory for Children (STAIC) test also contains anxiety, the relationship between anxiety and sleep disturbance measured in this study was interfered. I would like to suggest the authors delete the item score of sleep disturbance from the total score of anxiety and reanalyzed the data.

5.      The authors mentioned “upon the outbreak of the COVID19 pandemic, the number of children and adolescents presenting emotional and behavioral problems has significantly increased [17].” I would like to suggest the authors add the date of conducting this study and discussion regarding the potential influence of the COVID-19 pandemic on this study.

6.      Collecting data from parents only was a major limitation of this study. I would like to suggest the authors review the results of previous studies regarding the discrepancies of sleep disturbance and anxiety of children and adolescents between the reports of children and adolescents and parents and explain why collecting information for children and adolescents is important.

7.      Collecting the data from single sources of informants also raise biases. Please add discussion regarding it.

8.      The full spelling for the abbreviations such as BEARS and BRIEF-2 should be provided.

Author Response

Regarding the issues you raised,

  1.  “Cognitive flexibility” was one of variables examined in this study; however, it did not appear in the title.

We appreciate this comment. We wanted to highlight the emotional regulation, since we found more interesting results. But you are right, we have added it to the title. Thanks.

  1. The authors mentioned “the bidirectional correlation between sleep habits and emotional regulation.” The pathway of emotional regulation to sleep problems via anxiety was examined in this study. Why did the authors not consider the pathway of sleep problems to emotional regulation/cognitive flexibility via anxiety? According to the neuropsychological mechanism, this pathway can be hypothesized.

Thanks for this comment. We firstly aimed at studying the influence of emotional regulation/cognitive flexibility on sleep habits, since previous researchers mostly analyzed the other direction. The innovativeness of our research is to examine the effects of these dimensions of executive functioning on sleep habits, which have been hardly ever studied. However, correlations were initially included to the study and we have added other analyses based on your comments to justify the bidirectional relationship between these variables. 

  1. The reasons to examine the mediating role of anxiety for the association between emotional regulation/cognitive flexibility and sleep problems should be introduced.

We agree with your observation, and therefore, we have improved the explanation of the referenced concepts. Thank you very much for the correction.

  1. Sleep disturbance is often one of surveyed item in the instruments for anxiety. If the State-Trait Anxiety Inventory for Children (STAIC) test also contains anxiety, the relationship between anxiety and sleep disturbance measured in this study was interfered. I would like to suggest the authors delete the item score of sleep disturbance from the total score of anxiety and reanalyzed the data.

We appreciate this comment. We ran all the data analysis without the item you mentioned, but no significant differences were found. We therefore decided to keep all the initial items based on the factor structure of the questionnaire.

  1. The authors mentioned “upon the outbreak of the COVID19 pandemic, the number of children and adolescents presenting emotional and behavioral problems has significantly increased [17].” I would like to suggest the authors add the date of conducting this study and discussion regarding the potential influence of the COVID-19 pandemic on this study.

Correct, that fundamental piece of information had slipped our attention. The error has been rectified through the following explanation included in the Methodology: Participation in the study was requested through social media groups during the months of May and June 2022, using a snowball sampling strategy.

I would like to suggest the authors add the date of conducting this study and discussion regarding the potential influence of the COVID-19 pandemic on this study.

As you recommended, we have included an explanation about the possible effect that the COVID lockdown may have had on our sample. Thank you very much for the suggestion.

  1. Collecting data from parents only was a major limitation of this study. I would like to suggest the authors review the results of previous studies regarding the discrepancies of sleep disturbance and anxiety of children and adolescents between the reports of children and adolescents and parents and explain why collecting information for children and adolescents is important.

Thanks for this comment. You are right. Parent-report is the biggest limitation of our study. We have added some information related to self-rating in children/adolescents, according to your suggestions. 

  1. The full spelling for the abbreviations such as BEARS and BRIEF-2 should be provided.

Thanks for this appreciation. We have added the full spelling of these tests.

Reviewer 2 Report

The review of the manuscript entitled: “The influence of emotional regulation on sleep habits of Spanish children and adolescents through the lens of parents

The study aimed to analyze the correlations among sleep habits, anxiety and executive functions and the influence of emotional regulation and cognitive flexibility on sleep habits. The authors also tried to determine the mediating role of anxiety between executive functioning and sleep habits. For this purpose, 953 children and adolescents and their legal guardians was recruited. STAIC, BEARS and BREAF-2 was used to evaluate anxiety, sleep habits and executive behavior of the children and adolescents.

Comments for Authors:

Thank you for the research you have done. The paper is acceptably written and it is overall understandable. Methods and results sections are completely and clearly explained. However, there are some comments and questions:

1) STAIC questionnaire is a ‘self-report’ tool. Why authors manipulated the standard questionnaire to make it possible for guardians to answer the questions by proxy? However, some items of the scale are completely ‘subjective’ and others cannot answer them.

2) The study lacks novelty and the introduction and discussion sections cannot explain the necessity of the study.

Good luck

Author Response

Regarding the issues you raised,

  1. STAIC questionnaire is a ‘self-report’ tool. Why authors manipulated the standard questionnaire to make it possible for guardians to answer the questions by proxy? However, some items of the scale are completely ‘subjective’ and others cannot answer them.

Although the original instrument is a self-report one, the application was online. Messages were sent to families, who filled the surveys out in collaboration with children when necessary. However, we checked the reliability of the instrument for this study since we adapted the items. We indeed reported the reliability scores (see section 2.2. Instruments).

  1. The study lacks novelty and the introduction and discussion sections cannot explain the necessity of the study.

We firstly aimed at studying the influence of emotional regulation/cognitive flexibility on sleep habits, since previous researchers mostly analyzed the other direction. The innovativeness of our research is to examine the effects of these dimensions of executive functioning on sleep habits, which have been hardly ever studied. However, correlations were initially included to the study and we have added other analyses based on your comments to justify the bidirectional relationship between these variables and explain the need for the study in line with the introduction and discussion.

Reviewer 3 Report

The authors aim to analyze the influence of emotional regulation on sleep habits in Spanish children and adolescents. 953 Spanish parents completed the assessment according to their children and adolescents' information. the authors found moderate-strong correlations between emotional regulation problems and sleep habits disturbances, traits and state anxiety. Emotional regulation showed a direct impact on sleep habits. The authors conclude that emotional regulation may have an impact on sleep habits during childhood and adolescence.

Please justify the need for this study as it is known that emotional regulation will have an impact on sleep habits.

The questionnaires were filled out by parents. This may lead to parental bias. What measures were taken to address this?

the methodology appears confusing. I would suggest a framework to explain the flow of the study.

Author Response

Regarding the issues you raised,

  1. Please justify the need for this study as it is known that emotional regulation will have an impact on sleep habits.

The key in this study lies in the mediating role that executive functions play in emotional regulation and, therefore, in improving sleep habits despite the presence of anxiety. Nevertheless, we understand the need to explain it more clearly, and for this purpose, we have included some clarifying paragraphs in the introduction. We hope that after reading them, you will better understand the rationale for the study. Thank you for your guidance.

  1. The questionnaires were filled out by parents. This may lead to parental bias. What measures were taken to address this?

Thanks for this comment. You are right. Parent-report is the biggest limitation of our study. We have added some information related to self-rating in children/adolescents, highlighting the necessity of self reports. 

  1. The methodology appears confusing. I would suggest a framework to explain the flow of the study.

We have made some changes on the manuscript that we think they make the understanding of the methodology easier. Thanks for your comment.

Round 2

Reviewer 2 Report

The review of the revision of the manuscript entitled: “The influence of emotional regulation and cognitive flexibility on sleep habits of Spanish children and adolescents through the lens of parents

Comments for Authors:

Thanks for the revision you have done. However, there are some more issues:

1) ‘Introduction’ section, paragraph 8, lines 2-5, the authors mentioned: “In fact, studies show an overestimation of the parents when rating children’s behaviors, resulting in assessment biases. However, researchers have suggested that parent proxy forms are such an excellent candidate for accurately detecting cognitive and emotional aspects in youngsters”. I agree with both sentences. However, as mentioned before, STAIC questionnaire is a ‘self-report’ tool and some items of this scale are completely ‘objective’ and cannot be answer by parents. On the other words, manipulating the questionnaire by adding the statement of “your child feels…” before each item does not make a “parent proxy form” from it.

Moreover, authors have mentioned a new sentence in ‘methods’ section which is: “Although the original instrument was a self-report one, parents filled the online surveys out in collaboration with children with children when necessary”. Who can specify which item or sentence should be answered by the child necessarily? And who can guarantee that the result is exactly as accurate as the child answered the questions him/her self? Therefore, this would be a huge bias or it should be presumed that authors have used a questionnaire in their study, which is not exactly the STAIC scale.

Finally, the findings of this study cannot be expanded or compared with findings of other studies who has applied STAIC scale. This error is much bigger that can be addressed by mentioning it in the limitations paragraph of the manuscript.

2) Authors did not mentioned any inclusion/exclusion criteria for participants.

Good luck

Author Response

1) ‘Introduction’ section, paragraph 8, lines 2-5, the authors mentioned: “In fact, studies show an overestimation of the parents when rating children’s behaviors, resulting in assessment biases. However, researchers have suggested that parent proxy forms are such an excellent candidate for accurately detecting cognitive and emotional aspects in youngsters”. I agree with both sentences. However, as mentioned before, STAIC questionnaire is a ‘self-report’ tool and some items of this scale are completely ‘objective’ and cannot be answer by parents. On the other words, manipulating the questionnaire by adding the statement of “your child feels…” before each item does not make a “parent proxy form” from it. Moreover, authors have mentioned a new sentence in ‘methods’ section which is: “Although the original instrument was a self-report one, parents filled the online surveys out in collaboration with children with children when necessary”. Who can specify which item or sentence should be answered by the child necessarily? And who can guarantee that the result is exactly as accurate as the child answered the questions him/her self? Therefore, this would be a huge bias or it should be presumed that authors have used a questionnaire in their study, which is not exactly the STAIC scale. Finally, the findings of this study cannot be expanded or compared with findings of other studies who has applied STAIC scale. This error is much bigger that can be addressed by mentioning it in the limitations paragraph of the manuscript.

Thanks for your comments and suggestions. We have made some changes in the text based on your review. In particular, we have added new information in the following sections:

  1. Methods and Results. 
  2. Strength, limitations, and future research.

2) Authors did not mentioned any inclusion/exclusion criteria for participants.

We appreciate your insight. We have included it in section 2.1.: "Study population and design". Thanks. 

Round 3

Reviewer 2 Report

Thanks for the revision you have done. the manuscript is now much improved.

Good luck